# Suppression of Nasopharyngeal and Gastric Tumor Growth in a Mouse Model by Antibodies to Epstein–Barr Virus LMP1 Protein

**DOI:** 10.3390/microorganisms11071712

**Published:** 2023-06-30

**Authors:** Abdelhalim Khenchouche, Mounir M. Salem-Bekhit, Ahd A. Mansour, Mohammad N. Alomary, Xiaohui Wang, Hayat Ali Alzahrani, Ibrahim M. Al Hosiny, Ehab I. Taha, Gamal A. Shazly, Yacine Benguerba, Karim Houali

**Affiliations:** 1Département de Microbiologie, Faculté des Sciences de la Nature et de la Vie, Université Ferhat Abbas Sétif 1, Sétif 19000, Algeria; 2Laboratoire de Virologie Moléculaire, FRE3011, CNRS, Faculté de Médecine Laennec, Université Claude Bernard Lyon-1, 69008 Lyon, France; 3Department of Pharmaceutics, College of Pharmacy, King Saud University, P.O. Box 2457, Riyadh 11451, Saudi Arabia; 4Medical Laboratory Science Department, Fakeeh College for Medical Sciences, P.O. Box 2537, Jeddah 21461, Saudi Arabia; 5Advanced Diagnostic and Therapeutic Institute, King Abdulaziz City for Science and Technology, P.O. Box 6086, Riyadh 11442, Saudi Arabia; malomary@kacst.edu.sa; 6Department of Medical Laboratory Technology, Faculty of Applied Medical Sciences, Northern Border University, Arar 73211, Saudi Arabia; dr.hayatalzahrani@hotmail.com; 7Microbiology and Immunology Department, Faculty of Medicine, Al-Azhar University, Cairo 11651, Egypt; 8Laboratoire de Biopharmacie Et Pharmacotechnie (LPBT), Ferhat Abbas Setif 1 University, Setif 19000, Algeria; 9Laboratoire de Biochimie Analytique et Biotechnologie (LABAB), Faculté des Sciences Biologiques et des Sciences Agronomiques, Université Mouloud Mammeri, Tizi-Ouzou 15000, Algeria

**Keywords:** EBV oncogenes, LMP1, nasopharyngeal, gastric carcinomas, mouse model, tumor suppression and prevention

## Abstract

The study aimed to investigate the antitumor efficacy of anti-LMP1 antibodies in EBV-positive nasopharyngeal and stomach cell lines and xenograft models. The study also examined the NF-κB expression and cell cycle activation of NPC-serum-exosome-associated LMP1. Anti-LMP1 antibody treatment before or during cell implantation prevented tumor growth in nude mice. A small dose of antibodies resulted in complete tumor regression for at least three months after the tumors had grown in size. The consumption of antigen–antibody complexes by tumor cells limited tumor growth. In vitro experiments showed that anti-LMP1 antibodies killed EBV-positive NPC- or GC-derived epithelial cell lines and EBV-positive human B-cell lines but not EBV-negative cell lines. Treatment with anti-LMP1 reduced NF-κB expression in cells. The animal model experiments showed that anti-LMP1 inhibited and prevented NPC- or GC-derived tumor growth. The results suggest that LMP1 antibody immunotherapy could cure nasopharyngeal cancer, EBV-positive gastric carcinoma, and EBV-associated lymphomas. However, further validation of these findings is required through human clinical trials.

## 1. Introduction

The Epstein–Barr virus (EBV) is strongly linked to nasopharyngeal (NPC) and gastric carcinomas (GC), and other malignancies (Burkitt’s lymphoma and Hodgkin’s lymphoma); however, its causal status and role in genetic events are unknown. The EBERs, EBNA1, LMP1, LMP2A, BARF0, and BARF1 genes were all consistently expressed in NPC and GC biopsies. In rat fibroblasts, however, only LMP1 and BARF1 induced malignant transformation [1]. The mechanism of these two oncoproteins’ tumorigenic activity is not well understood. Latent Membrane Protein 1 (LMP1) and BamH1 A fragment Right Frame-1 (BARF1) are primarily expressed in NPC and EBV-related GC. LMP1, one of the essential genes involved in B-cell immortalization, activates a broad range of cell genes such as NF-κB, A20, and EGF-R [2].

According to recent research, LMP1, combined with exosomes, is abundantly released into the serum and saliva of NPC patients [3]. When isolated from NPC patient sera, LMP1/exosome complexes demonstrated a significant mitogenic response on cultured cells [4]. This suggests a role in tumor development via paracrine/exocrine action. Exosomal pathways are critical in the pathogenesis of EBV-related cancers. Understanding these pathogenesis-related mechanisms is essential for treatment and diagnosis. Exosomes were discovered to be secreted by Epstein–Barr virus (EBV)-infected cells for intercellular communication [5].

Latent EBV infections are important in the development of EBV-related cancer. Two latent membrane proteins (LMP1 and LMP2) are expressed during type III latency [6]. These viral proteins are expressed in the NPC and GC [7]. In this study, LMP1-specific antibodies were tested in nude mice implanted with NPC-derived (c666-1) or EBV-positive GC (AGS) tumor cells. Following the injection of the c666-1 cell line, LMP1 is secreted into the serum of these animals, producing NPC-type tumors, as seen in the serum and saliva of NPC patients [8].

The exosome-LMP1 isolated from the serum of NPC patients looks to have a significant role in tumor formation. The goal of our investigation was to see if the tumor might be stopped by targeting the circulating LMP1 oncoprotein with particular antibodies. Nude mice were injected with NPC-derived c666-1 or GC-derived AGS transformed into EBV-positive cell lines, which resulted in NPC-type or GC-type tumors. In the present study, we investigated whether anti-LMP1 antibodies can stop tumor growth in mice. Inactivation of the oncoprotein LMP1 may be sufficient to limit tumor development by suppressing NF-κB expression. It could then be used to treat nasopharyngeal cancer, EBV-positive gastric carcinoma, and/or EBV-associated lymphoma.

## 2. Materials and Methods

### 2.1. Antibodies

The S12 mouse monoclonal antibody that recognizes LMP1 was obtained from Becton Dickinson in France, while rabbit polyclonal anti-DNase is generated using the complete protein [9].

### 2.2. Cell Cultures

The c666-1 cells used in the study were cultured in RPMI 1640 medium supplemented with 10% fetal bovine serum (FBS) and antibiotics [10]. CEM (T cell) and B-lymphoid cells (EBV-positive AKATA, P3HR1, IB4, and Raji) were also cultured in RPMI 1640 medium supplemented with 10% FBS and antibiotics. Only the Raji EBV strain has the LMP1 sequence [10].

The EBV-AGS and AGS cell lines used in the study were obtained from Dr. Takada (Hokkaido University) and were grown in a DMEM medium. They were established in vitro by infecting cells with a recombinant EBV containing the neomycin resistance (Neor) gene at the BXLF1 site of EBV DNA [11].

### 2.3. Nude Mouse Experiments

At four weeks of age, Athymic Nude-Foxn1 mice were obtained from Harlan (France) and subcutaneously inoculated with either 10^7^ c666-1 or EBV-AGS cells. The mice were then given either experimental (S12) or control (anti-EBV DNase, anti-rabbit, or anti-mouse) antibodies intraperitoneally at a dose of 25 µg/injection based on the protocols presented in Figure 1 (protocols #1, 2, or 3). Tumor diameters were monitored daily, and the mice were anesthetized when their tumors grew beyond 2 cm in diameter. The mean values of eight mice were calculated. In an attempt to determine if the neutralization of the LMP1 oncoprotein by the specific antibody could prevent and/or inhibit tumor growth, anti-LMP1 antibodies were subcutaneously administered to nude mice that had been inoculated with 10^7^ cultured cells from EBV-associated tumors: c666-1 cells derived from NPC and EBV-AGS cells derived from GC.

### 2.4. Exosome/LMP1 Complex Preparation from NPC Patients’ Serum or Implanted Nude Mice

To isolate the LMP1/exosome complexes, we used a technique called differential centrifugation. The serum samples were subjected to ultracentrifugation at 2,000,000× *g* for 2 h, which allowed the exosomes to form a pellet. The latter was loaded onto a sucrose gradient ranging from 5% to 40% and centrifuged at 105,000× *g* for 16 h. The fractions that contained the LMP1/exosome complexes were identified using the S12 antibody in an immunoblotting assay, and the positive fractions were pooled together.

### 2.5. Immunoblotting Analysis

The proteins were isolated from tumor tissues using RIPA buffer containing 10% *w/v* of the tissue. The RIPA buffer contained 0.1% SDS, 0.5% Desoxycholate, 0.5% Triton-X100, 0.4 M NaCl, 5 mM EDTA, and 20 mM Tris-HCl at pH 7.6. Similarly, serum samples were prepared as described earlier. Next, 50 mg to 70 mg of protein were separated through 12% polyacrylamide gels using electrophoresis and transferred onto nitrocellulose membranes as previously described. The antigen–antibody complexes were detected using an improved chemiluminescence system known as ECL (ECL™ Western Blotting Detection Reagents), developed by Amersham [8].

### 2.6. Immunofluorescence Analysis

Tumor cells collected and processed to obtain single-cell suspensions were seeded and fixed with acetone onto glass coverslips, to preserve the cellular morphology and prevent protein degradation. The fixed cells are then permeabilized using Triton X-100 to allow access of the antibody to intracellular targets. Non-specific binding sites are blocked using bovine serum albumin (BSA) buffer. Anti-mouse Ig against S12 revealed the presence of the LMP1/exosome complex. For the exosome/LMP1 complex, tumor cells from c666-1 (as LMP1 c666-1) and EBV-AGS were used (as LMP1 EBV-AGS) as a positive control. Negative controls are c666-1 and EBV-AGS, which were not treated with anti-mouse Ig. The sample is then incubated with a secondary antibody conjugated to FITC, allowing visualization of the complex. To visualize the nuclei and cell membranes, the sample was counterstained with DAPI dye, then viewed under a fluorescence microscope.

### 2.7. Electron Microscopy Analysis

Exosomes isolated from mouse serum [12] were resuspended in PBS containing 0.02% sodium azide. They were fixed with 2% paraformaldehyde and dried on covered copper grids. The grids were then incubated with either anti-mouse Ig coupled to 20 nm gold beads or anti-CD63 linked to 10 nm gold beads. Raji cells treated with S12 antibody or Louckes cells treated with NPC-derived LMP1/exosome were cryoprotected in 2.3 M sucrose and frozen in liquid nitrogen before being fixed in 4% paraformaldehyde and 0.02% glutaraldehyde. Ultrathin sections were obtained using a dry microtome sectioning technique at −100 °C. The immunogold electron microscopy technique detected LMP1/exosome complexes obtained from mouse sera treated with S12 antibodies.

### 2.8. Confocal Microscopy Analysis

The LMP1/exosome complexes obtained from the sera of mice treated with S12 antibodies were visualized using anti-LMP1 or anti-CD63 antibodies, which are specific exosome markers. Louckes, CEM, and Raji cells were seeded in LAB-TEK II Chamber slide wells at 5 × 10^4^ cells/mL density and incubated with serum-free medium containing or without purified exosome/LMP1 from NPC serum. The cells were fixed in 4% paraformaldehyde and permeabilized with PBS containing 0.2% Triton X-100. DAPI fluorescent molecules were used to stain the nuclei. To label the cells, they were double-immunolabeled by first incubating them for two hours with the S12 antibody, followed by incubation for one hour with anti-mouse Alexa 633 (Rhodamine) (Invitrogen, France). The cells were then washed with PBS and incubated with anti-mouse anti-CD63 antibody, followed by anti-mouse Alexa 488 antibody (Green Fluorescein, Invitrogen, France).

### 2.9. Enzyme-Linked Immunosorbent Assay (ELISA) for NF-κB Components Detection

ELISA is a common laboratory technique used to detect and quantify protein samples. In this case, the TransAM NF-κB family kit was used to investigate the expression of five components of the NF-κB family (p65, p50, p52, RelB, and c-Rel) in the nuclei of tumor and cell homogenates. Cellular fractionation was performed to extract the nuclei. The homogenates were then lysed in a buffer containing various components and NP40 was added. The Optiprep nuclear pellet was lysed using the active Motif nuclear extraction buffer. The ELISA test was performed according to the manufacturer’s instructions. The expression of the NF-κB family components was analyzed in the nuclear extract by measuring the absorbance at 450 nm. Raji cells nuclear extract was used as a positive control for the antibody.

### 2.10. RT-PCR Analysis

In the study, two positive controls were used, P3HR1 and EBV + AKATA mRNA. To ensure the accuracy and reliability of the results, actin was used as a standard control. For the RT-PCR process, reverse transcription was performed, followed by quantitative PCR using the FastStart DNA Master SYBR green kit (Roche Diagnostics, Meylan, France) in a final volume of 20 µL. The primer sequences used in the study are provided in Table 1.

To determine the mRNA concentration, a calibration curve was utilized. The size of the amplified products was verified by gel electrophoresis using molecular weight markers during the initial PCR setting adjustment. Amplification specificity was checked using light cycler melting-curve analysis once the PCR product size was confirmed. The Light Cycler quantification software 4.1 quantified the transcripts in samples, and the data were calibrated from repeated dilutions of purified PCR products containing known amounts of cDNA molecules from each gene (Roche Diagnostics). The ratio of LMP1 mRNA to actin mRNA was calculated to determine the relative abundance of each mRNA. This experiment was replicated three times.
$$\mathrm{Normalized}\;\mathrm{ratio}={\left(\frac{\mathrm{conc}.\;\mathrm{target}}{\mathrm{conc}.\;\mathrm{reference}}\right)}_{\mathrm{sample}}:{\left(\frac{\mathrm{conc}.\;\mathrm{target}}{\mathrm{conc}.\;\mathrm{reference}}\right)}_{\mathrm{calibrator}}$$

### 2.11. Proliferation Analysis

In this experiment, 5 µL of the exosome/LMP1 complex from an NPC patient serum (SNPC) was used in a culture medium without FBS to test cell proliferation, as described previously [13]. The controls used were exosomes isolated from healthy individuals (EC-SNF), with or without FBS. To determine cell proliferation and viability, the MTT assay (Cell Proliferation Kit I; Roche) was used, as previously reported [12]. Cells were seeded at 10^4^ cells per well in 96-well culture plates, and when they reached 60–70% confluence, 10 µL of serum-free RPMI 1640 medium containing 300 ng of exosome/LMP1 complex was added to each well. After 48 h of culture, 10 µL of MTT solution was added, and the cells were incubated for an additional four hours. The formazan was dissolved overnight at 37 °C in 100 µL of solubilization solution, and the optical density was measured at 450 nm. As we talk about proliferation, we will convert it into a measurable variable using the following formula:$$\mathrm{Proliferation}\;\mathrm{Rate}\;\left(\%\right)=\frac{\mathrm{Absorbance}\;\mathrm{of}\;\mathrm{treated}\;\mathrm{sample}-\mathrm{Absorbance}\;\mathrm{of}\;\mathrm{blanc}}{\mathrm{Absorbance}\;\mathrm{of}\;\mathrm{control}-\mathrm{Absorbance}\;\mathrm{of}\;\mathrm{blanc}}\times 100$$

### 2.12. Statistical Analysis

The statistical analysis for the tests involved the use of the Mann–Whitney HSD test with a significance level of *p* < 0.05. The tests were performed in triplicate, and the results were presented as mean standard deviation after one-way ANOVA and Tukey’s HSD test with a significance level of *p* < 0.05. The statistical software used for all analyses was SPSS Statistics, version 25.0.

## 3. Results

### 3.1. LMP1 Antibodies’ Impact on Tumor Growth

A previous study [8] found that EBV-LMP1 oncoprotein isolated from NPC patients’ sera had strong mitogenic activity in vitro. In this present study, mice were left untreated and tumors were observed to grow rapidly, reaching a diameter of 4 mm by day 4, 7–8 mm (c666-1) or 12–15 mm (AGS) by day 8, and 1.5–1.6 cm (c666-1) or 1.8–2 cm (AGS) by day 14 (Figure 1a and Figure 2). EBV-AGS tumors were significantly larger than c666-1 tumors. In a preventive protocol (protocol #1), antibodies against LMP1 (S12) were administered as five intraperitoneal injections of 25 µg at 5-day intervals, finishing three days before tumor growth, or as five consecutive daily injections beginning either simultaneously with tumor development (protocol #2) or on day 8 when the tumors had reached a diameter of approximately 8 mm (c666-1) or 12–15 mm (AGS) (protocol #3). After five daily injections of anti-LMP1 antibody, nodules of approximately 8 mm (c666-1) or approximately 12–15 mm (AGS) rapidly stabilized and then regressed progressively (Figure 1d and Figure 3). The tumor masses vanished completely 11 days after treatment began, and the mice remained tumor-free for at least three months. There were no significant differences between the control (non-treated animals) and those treated with anti-EBV DNase or anti-rabbit immunoglobulins. The study, however, revealed a significant difference between the control (non-treated animals) and those treated with S12 antibodies (anti-LMP1) as a preventive or post-tumor growth LMP1 antibody treatment.

LMP1 S12 is an antibody derived entirely from mice, with sodium azide making up 0.05% of its composition manufactured by Becton Dickinson in France. It does not trigger apoptosis, and there is no evidence of apoptotic bodies being observed. Instead, it operates through a cellular immune response mechanism. Although the precise details of this mechanism remain unknown, it is clear that apoptosis is not triggered.

### 3.2. Presence of Exosome/LMP1 and Exosome/LMP1/S12 Complex in Mouse Serum and Tumor Cells

Differential centrifugation is a technique used to separate particles based on their size and density. In this case, it was used to separate serum components to isolate the LMP1/exosome complexes. he western blot analysis is used to detect specific proteins in the samples. The proteins are sorted by size using 12% SDS-polyacrylamide gel electrophoresis before being transferred to a membrane. After that, the membrane is probed with particular antibodies to detect the target protein (Figure 4). In this case, anti-LMP1 and anti-mouse Ig antibodies were used to detect the LMP1/exosome complexes in the serum and tumor biopsy samples. The complexes associated with mouse immunoglobulin indicate that the S12 antibody could bind to and target the LMP1/exosome complexes.

The presence of LMP1/exosome complexes was confirmed using immunogold electron microscopy (Figure 5). The analysis was performed on LMP1/exosome complexes isolated from the serum of mice that had been treated with anti-LMP1 antibodies. The presence of exosomes containing the LMP1-S12 complex was confirmed through a positive response to anti-mouse Ig reacting with S12 antibody and anti-CD63 (Figure 3 + anti-CD63). A negative response was observed in exosomes isolated from mouse serum that had not been treated with anti-LMP1 (Figure 3, normal exosome).

The LMP1/exosome/S12 antibody complexes were discovered inside cells isolated from tumor biopsies of treated mice, as illustrated in Figure 4 (tumor cells + anti-Ig). The tumor cells were extracted from the biopsy, mounted on a slide, and then treated with anti-mouse Ig to reveal the LMP1/S12 complex. LMP1/exosome/S12 complexes were observed at the tumor cell plasma membrane, and significant patchy accumulations in the cytoplasm in both c666-1 and EBV-AGS tumor cells (LMP1 c666-1 and LMP1 EBV-AGS). Nuclei fluorescence was also revealed. Anti-Ig antibodies were tested as negative controls against mouse tumor cells extracted from tumor-induced mice not treated with LMP1 antibody. No significant response was observed (Figure 6, negative control). The normally mitogenic component was rendered ineffective by combining it with its specific antibody.

### 3.3. Combination of Confocal Microscopy, Immunoelectron Microscopy, and Immunofluorescence

After incubating LMP1/exosome complexes with CEM T-cell and Louckes B-cell lines, the cellular localization of the complex was examined using confocal microscopy. The results showed co-localization of LMP1 and CD63 in the cellular membrane and extensive cytoplasmic, and perinuclear patches in both cell lines (Figure 7). In CEM T cells, the complex was also observed in the nucleus (Figure 7: CEM1 + exosome LMP1).

Confocal microscopy was used to identify the exosome/LMP1 complex and CD63 in exosome/LMP1-treated Louckes, CEM, and Raji cells. Raji cells were treated with S12 for 24 h, while CEM and Louckes were treated with an exosome/LMP1 complex purified from an NPC patient’s serum. The cells were fixed and stained with DAPI to distinguish the nucleus. After incubation with S12 or anti-CD63 antibodies, followed by incubation with Alexa fluor 488 IgG goat anti-mouse IgG as a secondary antibody, LMP1 fluoresced in red with rhodamine, and CD63 fluoresced in green with fluorescein. The cells were excited at 356 and 488 nm (DAPI) (Alexa). At the 96 h interval of treatment, we observed a similar nuclear localization in Raji cells treated with S12 (anti-LMP1 antibody) (Figure 8). When Raji cells were exposed to mouse Ig, there was no fluorescence (Raji + Mouse Ig).

Immunogold electron microscopy study showed that Louckes cells treated with LMP1/exosome complexes incorporated the complex into a large multilobular intracytoplasmic vesicle containing both LMP1 and CD63 (Figure 9a–c). S12-antibody-treated Raji cells also exhibited a similar localization of the complex, indicating that the antibody did not affect the complex’s location (Figure 9d–f). The complex was also observed in the nucleus (Figure 9b) and as multimeric exosomes in the extracellular compartment (Figure 9a,d).

### 3.4. Mitogenic Stimulation of Several Cell Lines by LMP1/Exosome Extracted from NPC Serum and Its Deactivation by Anti-LMP1 Antibody

The proliferation activity of the LMP1/exosome complex was measured in several EBV-negative cell lines, including human T CES, human B AKATA (EBV- variant), rodent fibroblast Balb/c3T3, and human epithelial HaCaT, as shown in Figure 10. The fetal bovine serum (FBS) was replaced with the LMP1/exosome purified from NPC serum, and the MTT test was used to measure the proliferation activity.

The data obtained from the MTT test was analyzed using the Mann–Whitney HSD test at a significance level of 0.05. This analysis aimed to explore variations in cell cycle activity among different cell lines. The assays were conducted three times, and the results showed no significant differences between the cell lines treated with EC(SNP), ELC(SNPC) + S12, and FBS (*p* = 0.05). Likewise, there were no significant variations observed between the cell lines treated with 10% FBS and ELC(SNPC). However, the statistical analysis revealed a significant difference between the group comprising FBS-, EC(SNP), and ELC(SNPC) + S12, and the group consisting of FBS 10% and ELC(SNPC).

### 3.5. Effect of S12 Antibodies on Different Cell Lines

The effect of LMP1 antibodies on various cell lines, including Louckes, IB4, Raji lymphoblastoid cells, c666-1, AGS, and EBV-AGS epithelial lines, was tested. The growth of the Raji and IB4 cell lines was inhibited by the S12 antibody after treatment with 5 µg, with only 1% of viable cells normally developing (as shown in Figure 11a,d). The anti-LMP1 antibody was found to affect the c666-1 cell line (as indicated in Figure 11c), while the Louckes and AGS cells were not affected by it (as shown in Figure 11b,e, respectively). The Louckes and EBV-AGS cell lines were slightly inhibited by the antibody (as shown in Figure 11c,f).

The Mann–Whitney Honestly Significant Difference (HSD) test was used to examine the effect of S12 antibodies on the survival of different cell lines at *p* ≤ 0.05. The anti-LMP1 antibody did not affect the Louckes and AGS lymphoblastoid lines, and there were no significant changes between them or between growth with and without S12.

### 3.6. LMP1 Transcriptional Expression in Cell Lines and Tumors

Quantitative RT-PCR was used to compare the relative transcriptional expression of LMP1 in various cells, including P3HR1 (as a positive control), EBV-positive AKATA, c666-1, AGS, and EBV-AGS cells, both ex vivo and in culture (as shown in Figure 12). The results indicated that P3HR1 cells transcribed significant mRNA. In contrast, EBV-AKATA cells did not show any important transcription. We observed that LMP1 expression was absent in both AGS-EBV cell cultures and AGS cells. However, LMP1 expression became positive in tumor cells. In the case of c666-1 cells, a significant increase in LMP1 transcription was observed in the tumor.

EBV-P3HR1 mRNA was used as a positive control. Actin was used as a standard control. Real-time RT-PCR on the Light Cycler (Roche Diagnostics, Barrington, IL, USA) was performed. Samples were quantified using the Light Cycler relative quantification software 4.1 (Roche Diagnostics). Specific mRNA levels were expressed as the LMP1 mRNA/actin mRNA ratio percentage. All experiments were performed in triplicate.

### 3.7. Translational Expression of NF-κB

It was found that LMP1 promoted NF-κB expression, as previously reported by Kieff and Rickinson [14]. We also looked into the expression of various NF-κB family members, including p65, p50, p52, RelB, and c-Rel. Treatment with the S12 antibody completely suppressed the expression of NF-κB p65 and p50, which are essential components of the NF-κB family, in Raji and c666-1 cells 24 h after treatment (as shown in Figure 13). This suggests that the expression of NF-κB p65 and p50 in these cells is entirely dependent on LMP1 activation.

In biopsy samples obtained from untreated mice implanted with c666-1 or EBV-AGS cells, the expression of NF-κB, p65, and p50 was highly activated (as shown in Figure 13a, labeled as EBV-AGS Tumor and c666-1 Tumor). Moreover, Louckes cultures treated with ELC (exosome/LMP1 complex from NPC serum) exhibited an increase in p65 and p50 expression (as shown in Figure 13b, labeled as Louckes + ELC).

In Figure 13a, analysis was conducted on the expression of the NF-κB family in different cell lines including AGS, EBV-AGS, S12-treated EBV-AGS, EBV-AGS tumor, c666-1, S12-treated c666-1, and c666-1 tumor. It was found that the major components of the NF-κB family, p65 and p50, were mainly expressed in c666-1 cells, while EBV-AGS cells did not express significant amounts of p65 and p50 due to the absence of LMP1 gene expression. The statistical analysis with the Honestly Significant Difference test (HSD) from Mann–Whitney at *p* < 0.05 revealed that EBV-AGS, AGS, EBV-AGS-Sde p65, and p50 cells had very low expression. The anti-LMP1 antibody did not affect the Louckes and AGS lymphoblastoid lines; there were no changes between them and C666-1-S12. However, the expression of p65 and p50 differed significantly from that of other NF-B family proteins. The expression of NF-κB, p65, and p50 was significantly inhibited in S12-treated c666-1 cells. Without this antibody, these two major components’ expression was activated in tumors induced by both EBV-AGS and c666-1 cells.

In Figure 13b, analysis was conducted on the expression of the NF-κB family in different cell lines, including Louckes, Louckes + ELC (Exosome/LMP1 Complex, prepared from NPC serum), Louckes + S12, Louckes + ELC + S12, Raji, and S12-treated Raji. It was found that the expression of two major components, p65 and p50, was induced in Louckes cells by ELC, and their expression was significantly inhibited by the presence of anti-LMP1 S12 antibody (Louckes + ELC + S12). In Raji cells, p65 and p50 were found to be highly expressed, while other components (p52, RelB, and c-Rel) were slightly expressed. The expression of the NF-κB family was found to be diminished in the presence of the S12 antibody (Raji + S12).

## 4. Discussion

It was previously found that exosome-like vesicles in the serum of NPC patients were associated with LMP1 [8]. LMP1 is primarily located in intracytoplasmic lipid rafts and perinuclear accumulations [14]. Exosome-associated LMP1 was then investigated in mice with NPC or GC tumors. The LMP1 oncoprotein was present in the serum of mice with c666-1 (NPC) and EBV-AGS (GC) tumors, and LMP1-specific antibodies inhibited tumor growth both in vitro and in vivo. LMP1/exosome complexes could be secreted by all EBV-positive B-cell and epithelial cell lines and were involved in NF-κB activation in most cells. The addition of the S12 antibody in the medium was sufficient to inhibit Raji cell growth, indicating that the normal secretion of LMP1 complexed with exosomes functions as an autocrine/paracrine mechanism for cellular proliferation [15]. LMP1 and exosomes complexed with antibodies were identified in perinuclear and intranuclear patches of the inhibited cells.

LMP1/exosome complex, when added to EBV-negative cell lines, activates the cell cycle; it is for this reason that we have synonymously called it “mitogenic activity”. This suggests that the presence of the LMP1/exosome complex stimulates cellular processes leading to cell division. However, when produced by malignant cells, the LMP1/exosome complex can lead to immune evasion and cell proliferation, as reported by Aga et al. [16] and Meckes and Traub [17]. Fetal calf serum, which is commonly used as a cell culture supplement, also activates the cell cycle. The inclusion of fetal calf serum in the culture medium provides essential nutrients and growth factors that promote cell proliferation and entry into the cell cycle. Interestingly, when the fetal calf serum is replaced by the LMP1/exosome complex, it still activates the cell cycle, specifically mitosis. This suggests that the LMP1/exosome complex is able to mimic the effects of fetal calf serum in promoting cell cycle progression in the absence of growth factors.

When the LMP1/exosome complex is combined with anti-LMP1 antibodies, it results in the arrest of the cell cycle. This indicates that the presence of anti-LMP1 antibodies counteracts the stimulatory effects of the LMP1/exosome complex, leading to cell cycle arrest. This finding suggests that the anti-LMP1 antibodies have the potential to interfere with the signaling pathways involved in cell cycle regulation. These findings highlight the importance of LMP1 and exosome interactions in cell cycle regulation and suggest a potential role for LMP1/exosome-complex-mediated signaling in promoting cell division. The ability of anti-LMP1 antibodies to arrest the cell cycle further underscores the potential therapeutic implications of targeting LMP1-associated pathways in diseases where LMP1 is implicated, such as Epstein–Barr virus (EBV)-associated malignancies.

Several cell lines, including AKATA, Louckes, CEM, Balb/c3T3, and HaCaT, were used in the study. The exosome/LMP1 complex (ELC) was able to develop tumors in mice. Adding S12 to the exosome/LMP1 assay eliminated almost all the cell proliferation activity (ELC + S12). The inhibition of this proliferation resulted from the inactivation of LMP1 localized on the exosomes by S12, the anti-LMP1 antibody. The mitogenic activity is caused by the exosome/LMP1 complex circulating in the serum of NPC patients. According to Chen et al. [18], the key pathogenic mechanisms of the virus may not be the viral particles themselves but the exosome, which plays a critical role in EBV malignancies. Exosomes serve as ideal carriers for LMP1 molecules, preventing them from being degraded by host enzymes and transporting them to other cells under the constant influence of target cells [19].

LMP1 was found to activate FGF2 expression when found in exosome-like vesicles [20]. The authors hypothesized that exosome-associated LMP1 may react with the S12 antibody in the extracellular medium before reincorporating into a large intracytoplasmic endosome and redistributing to the cell membrane or secreting. LMP1 is found in extracellular membrane-derived vesicles during EBV infection [21,22]. A significant reduction in NF-κB expression was observed in cells treated in vitro with the S12 antibody. The absence of NF-κB, p65, and p50 could cause cellular apoptosis and tumor suppression [23], which could be triggered by exosome/LMP1/S12 complexes in the cytoplasm and nuclei.

The mitogenic activity was previously observed in EBV-negative cells that were exposed to the LMP1/exosome complex isolated from the serum of patients with NPC [24]. However, when the free form of LMP1 (without exosome) and the free exosome (without LMP1) were isolated from a normal patient’s serum, no proliferation was observed [12]. These findings suggest that the complexation of LMP1 and exosome is essential for mitogenic activity. Exosomes are capable of being efficiently secreted and absorbed by cells through simple membrane fusion [5,12,25] and can potentially enter the cell via endocytosis [26]. Once incorporated into endosomes, the LMP1 within the exosome can activate cellular proteins, such as NF-κB, just before being secreted outside of the cells [27].

It is possible that the exosome, associated with LMP1, acts as a vehicle to facilitate absorption into cells via endocytosis, allowing LMP1 to enter and become active with the help of the exosome. However, further research is necessary to determine whether exosomes isolated from NPC serum have mitogenic activity [27]. The S12 antibody was able to inactivate exosome/LMP1 complexes after they were secreted, which greatly reduced the activation of cellular proteins, implying that LMP1 on the exosome is required for NF-κB activation.

Incorporating LMP1-exosome complexes into cells may interact with the TRAF family and TRADD in CTAR regions to activate NF-κB and/or AP-1 [28]. The C-terminal sequence of LMP1 appears to be accessible to these cellular proteins when complexed [29]. Nuclear ELC may also activate other nuclear proteins involved in cell proliferation. The findings on ELC are similar to the observed phenomenon with the EGFR molecule, which was found to activate NF-κB and bind to the promoter regions of cyclin D1 and cyclin E [30,31].

Cell death was observed when S12 was added to the Raji cell culture, and the ELC/S12 complex was found in the nucleus. The results confirm that LMP1 complexed with exosomes is required for cell cycle entry. The exosome/LMP1 complex may regulate NF-κB expression, as only the S12 antibody inhibited NF-κB expression in EBV-positive cell cultures [32]. Patients with NPC have insufficient antibody responses to this viral protein, and immunotherapy targeting LMP1 could effectively prevent and treat NPC [33]. Other EBV-associated tumors may also be cured, as the BL-derived cell lines studied were all susceptible to anti-LMP1 antibodies. Further research is necessary to clarify the interaction of ELC with vimentin and/or importin alpha/ß1 in nuclear importation [34].

LMP1 is a transmembrane protein that is expressed in EBV-infected cells. It has been shown to have a variety of effects on cellular processes, including the cell cycle, apoptosis, and immune signaling. LMP1 is thought to play a key role in the development of EBV-associated cancers by promoting cell proliferation and inhibiting apoptosis [2].

Research has shown that LMP1 is complexed with exosomes in the serum of NPC patients, and this complex has mitogenic activity in EBV-negative cells. The complex can enter cells via endocytosis, where LMP1 can activate cellular proteins such as NF-κB. When anti-LMP1 antibodies neutralize LMP1, NF-κB expression is greatly reduced. These findings suggest that the exosome/LMP1 complex is required for cell cycle entry and the regulation of NF-κB expression.

The involvement of LMP1 in gastric carcinoma is also worth considering [35]. LMP1 has been identified as a potential biomarker for GC associated with EBV diagnosis and prognosis, especially when complexed with exosomes. It is thought that a decrease or increase in plasma EBV DNA levels may correlate with the clinical course, which could be useful in clinical practice [36].

## 5. Conclusions

The Epstein–Barr virus (EBV) is a common virus that infects most people at some point in their lives. While EBV is generally harmless, it has been associated with several types of cancer, including nasopharyngeal carcinoma (NPC) and gastric carcinoma (GC). One of the viral proteins that has been implicated in these cancers is LMP1.

Immunotherapy targeting LMP1 has shown promise for treating and preventing NPC and other EBV-associated tumors. This approach involves the use of antibodies that target LMP1 and block its activity, leading to cell death or the immune-mediated clearance of infected cells. Since patients with EBV-related cancers often have insufficient antibody responses to LMP1, developing targeted therapies is a promising avenue for treatment.

In conclusion, the research presented provides valuable insights into diagnosing and treating EBV-related cancers. The involvement of LMP1 in cell cycle regulation and its potential as a biomarker for GC diagnosis and prognosis is worth considering. Developing targeted therapies that can neutralize LMP1 and block its activity is promising for treating and preventing NPC, GC, and other EBV-associated tumors. Further research in this area could lead to new treatment options for patients with these types of cancers.

## Figures and Tables

**Figure 1 microorganisms-11-01712-f001:**
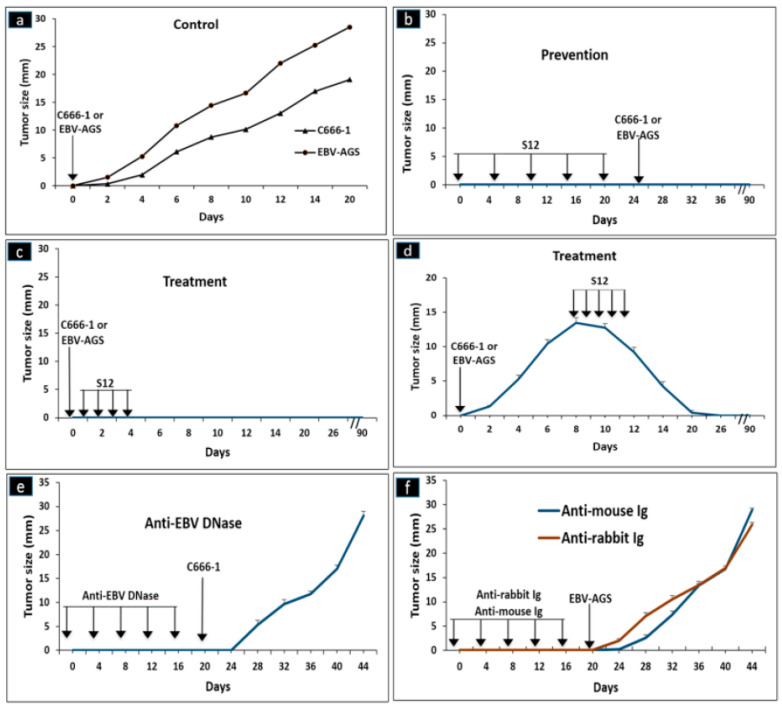
Immunotherapy assays: As a negative control, C666-1 or EBV-AGS cells are injected without treatment with S12 (protocol: **a**). Anti-LMP1 S12 was injected before (prevention protocol: **b**), concurrently with (treatment protocol: **c**), or following the injection of c666-1 or EBV-AGS cells (treatment protocol: **d**). As controls, anti-EBV DNase (**e**), anti-mouse Ig, or anti-rabbit Ig (**f**) were injected in place of S12. Each antibody was injected intraperitoneally at a dose of 50 µg. Subcutaneously, 10^7^ cells (c666-1 or EBV-AGS) were injected. Eight mice were used in each experiment, and the values in the figure correspond to the average tumor diameter in millimeters.

**Figure 2 microorganisms-11-01712-f002:**
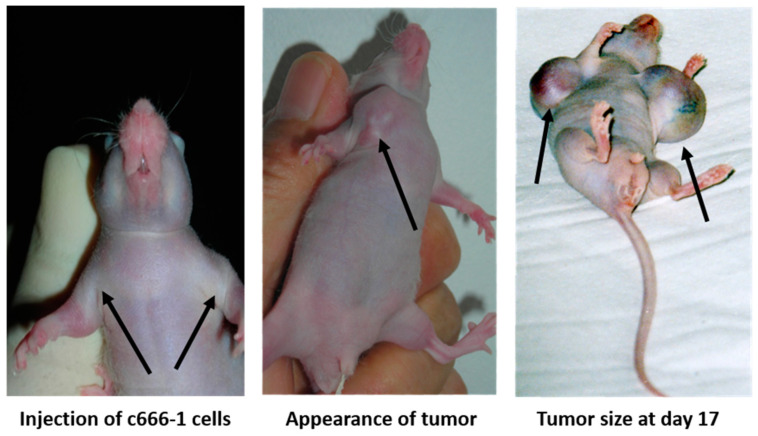
Tumor induction in a nude mouse by cC666-1 injection.

**Figure 3 microorganisms-11-01712-f003:**
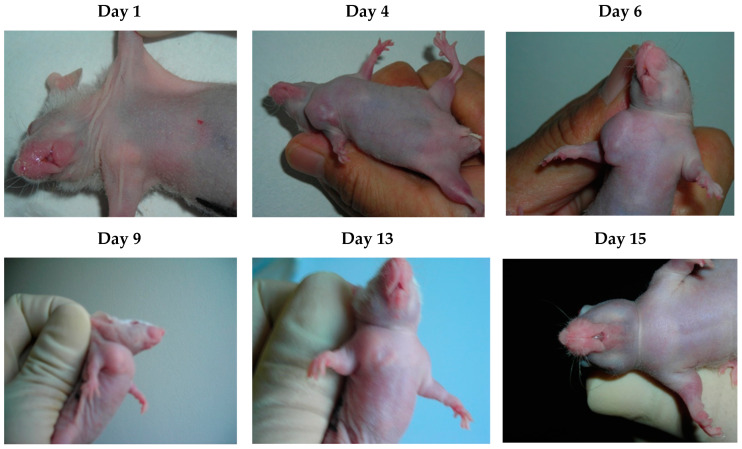
Tumor induction and therapy with S12 antibodies.

**Figure 4 microorganisms-11-01712-f004:**
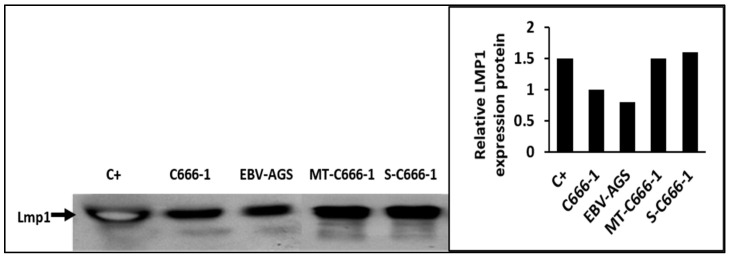
Detection of exosome/LMP1/S12 complex in serum and tumor by western blot. LMP1 was found in serum from mice developing c666-1, or EBV-AGS tumors. P3HR1 was a positive control (C+) for LMP1. S-c666-1: serum-extracted LMP1/exosome complex from c666-1 tumor-bearing animals. MT-c666-1: LMP1/exosome/S12 complex isolated from c666-1 tumor.

**Figure 5 microorganisms-11-01712-f005:**
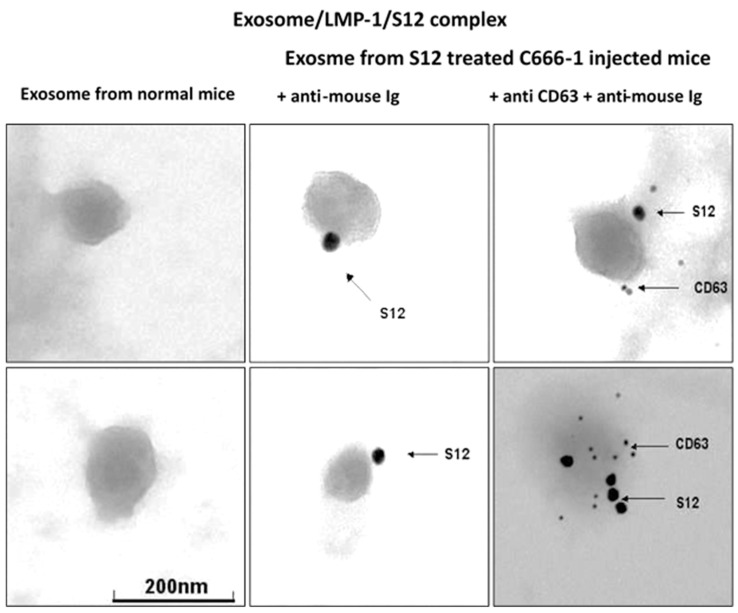
Detection of exosome/LMP1/S12 complex in serum by electron microscopy.

**Figure 6 microorganisms-11-01712-f006:**
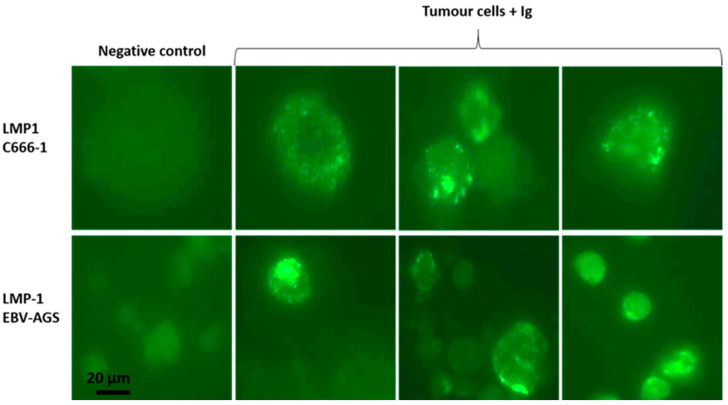
Detection of exosome/LMP1/S12 complex in tumor cells by immunofluorescence: Tumor cells were laid on a slide and fixed with acetone. Anti-mouse Ig for S12 revealed the presence of the complex. For the exosome/LMP1 complex, tumor cells from c666-1 (as LMP1 c666-1) and EBV-AGS were used (as LMP1 EBV-AGS). Negative controls are c666-1 and EBV-AGS, which were not treated with anti-mouse Ig.

**Figure 7 microorganisms-11-01712-f007:**
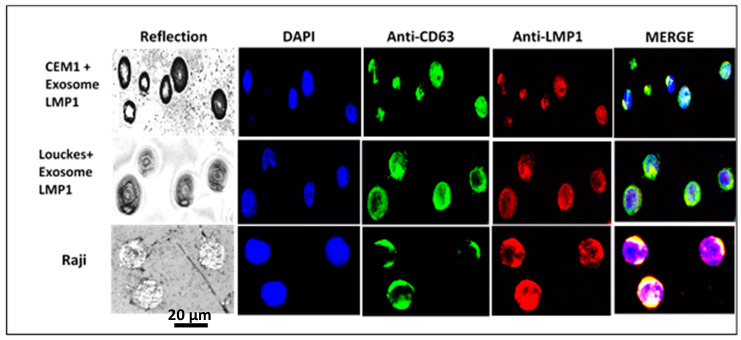
Identification of exosome/LMP1 complex in NPC-derived exosome/LMP1-treated B and T cells, and Raji cells by confocal microscopy, Immunoelectron microscopy and immunofluorescence were used to identify the exosome/LMP1/S12 complex.

**Figure 8 microorganisms-11-01712-f008:**
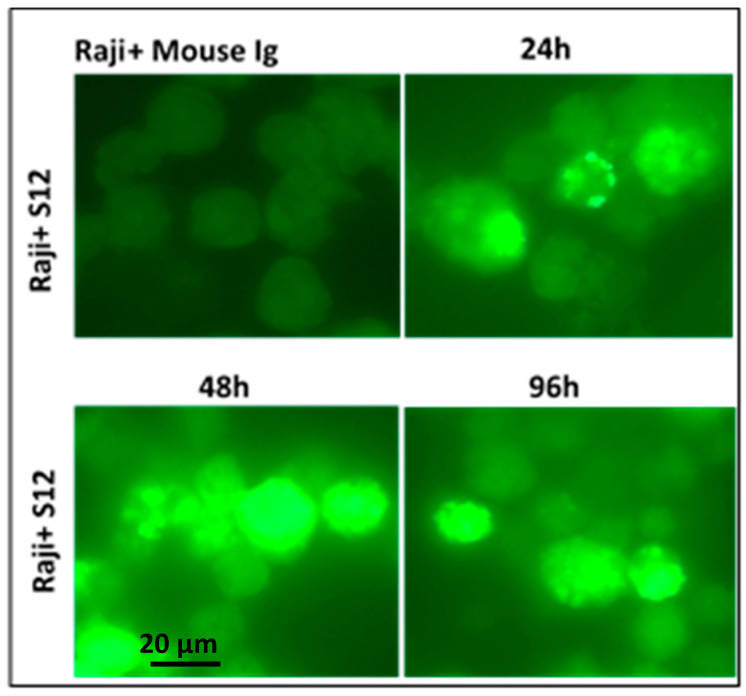
Identification of exosome/LMP1/S12 complex by immunofluorescence: Raji cells were cultured for 96 h in the presence of 5 g S12. Anti-mouse Ig-coupled with fluorescein was used to detect the exosome/LMP1/S12 complex. Control Raji cells were incubated with mouse Ig before being reacted with fluorescein-conjugated anti-mouse Ig (Raji + Mouse Ig). The cells had no fluorescence, indicating that mouse Ig was not absorbed.

**Figure 9 microorganisms-11-01712-f009:**
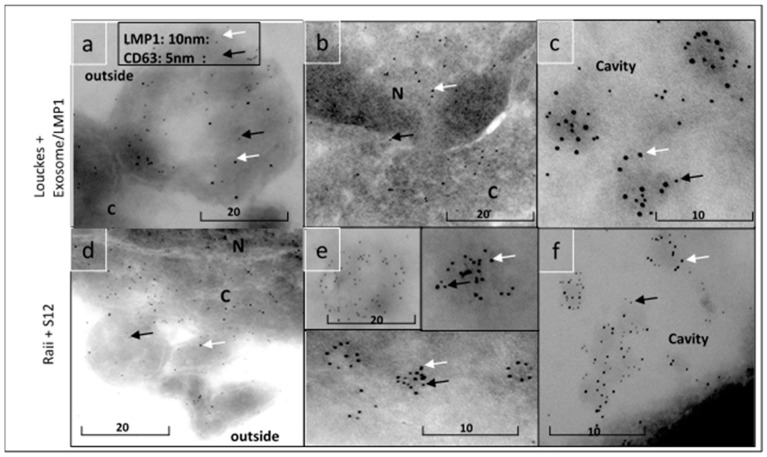
Identification of exosome/LMP1/S12 complex by immunoelectron microscopy: For 48 h, Louckes cells were treated with serum containing an exosome/LMP1 complex purified from NPC. Cell pellets were cut while frozen and placed on a slide. An anti-CD63 antibody was used in conjunction with a 10 nm gold bead to detect CD63. S12 coupled with a 5 nm gold bead detected LMP1. Raji cells were fixed after 48 h of treatment with the S12 antibody. Anti-mouse Ig (for S12) or anti-CD63 antibodies were used to treat the slides (for exosome). Anti-mouse Ig (coupled with a 5 nm gold bead in black allow) reacts to S12 antibody on exosome/LM1/S12 complex, and anti-CD63 (coupled with a 10 nm gold bead in white allow) reacts to CD63 on exosome. Exosomal vesicle (**a**,**c**,**d**–**f**), multi-vesicle (**a**,**d**), cavity (**c**,**f**) and nuclei (**b**).

**Figure 10 microorganisms-11-01712-f010:**
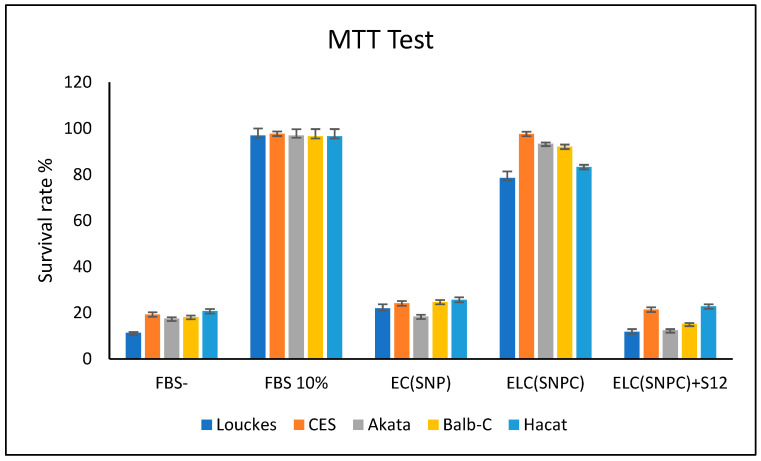
MTT test for CES, EBV-negative AKATA, Balb/c3T3, and HaCaT treated with exosome/LMP1 isolated from the serum of NPC patients.

**Figure 11 microorganisms-11-01712-f011:**
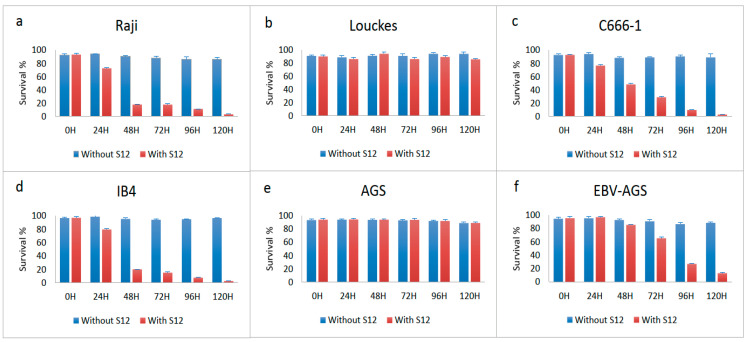
Effect of S12 on c666-1 and EBV-AGS cell growth: The effect of S12 was tested on Raji (**a**), Louckes (**b**), c666-1 (**c**), IB4 (**d**), AGS (**e**), and EBV-AGS (**f**) cell growth.

**Figure 12 microorganisms-11-01712-f012:**
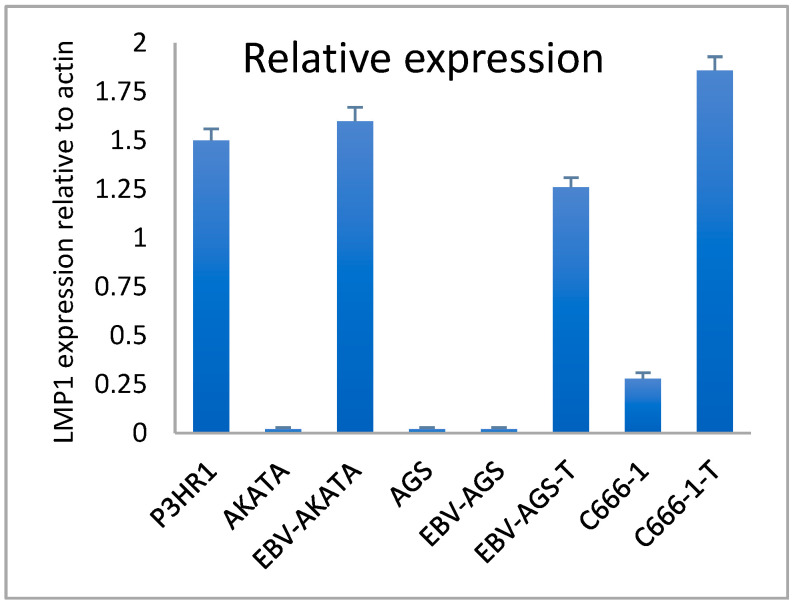
Transcriptional expression of LMP1 in c666-1 and EBV-AGS cells by RT-PCR (data quantified by the relative method to actin).

**Figure 13 microorganisms-11-01712-f013:**
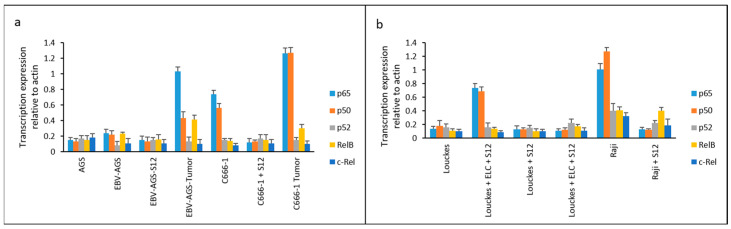
Translational expression of NF-κB family in c666-1, AGS, EBV-AGS, c666-1 tumors, and EBV-AGS tumors. (**a**) labeled as EBV-AGS Tumor and c666-1 Tumor, (**b**) labeled as Louckes + ELC.

**Table 1 microorganisms-11-01712-t001:** Primers used in RT-PCR assay to quantify EBV-LMP1.

	Actin Primers	LMP1 Primers (Exon 3)
Sens	5′-CCTTCCTGGGCATGGAGTCCT-3′	5′-CGGGATCCATGGAACGCGACCTTGAGAG-3′
Antisense	5′-GGAGCAATGATCTTGATCTTC-3′	5′-CGGAATTCTAAGCAGGATGATGGCTAGG-3′

## Data Availability

Not applicable.

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
