# Peer review of "Suppression of Nasopharyngeal and Gastric Tumor Growth in a Mouse Model by Antibodies to Epstein–Barr Virus LMP1 Protein"

_microorganisms, 2023, doi:10.3390/microorganisms11071712_

Round 1

Reviewer 1 Report (Previous Reviewer 3)

This is interesting paper and now it is much improved than previous version. So, now it is acceptable.

Please amend title "Suppression of Nasopharyngeal and Gastric Tumors Growth" like "Suppression of Nasopharyngeal and Gastric Tumor Growth"

Reviewer 2 Report (Previous Reviewer 2)

None

This manuscript is a resubmission of an earlier submission. The following is a list of the peer review reports and author responses from that submission.

Round 1

Reviewer 1 Report

Authors should consider the following comments to improve their manuscript:

1. References should be updated; all references cited in the manuscript are from 1985 to 2007.

2. Use appropriate citations in both introduction and discussion, specially to support results.

3. Materials and Methods section need to be improved, showing more details about the experimental design.

4. Figures 1, 5 and 7 are too crowded.

5. Results section must be improved; it is too hard to follow. Authors mentioned the use of an antibody to control tumor growth but the amount used in their experiments still unknown.

6. If two cancer cell lines are considered LMP1 active, then why to include other analysis that are unrelated to this study.

Author Response

Dear Reviewer,

Thank you for taking the time to review our manuscript and for your valuable feedback. We have carefully considered all of your comments and have made the following revisions:

References should be updated; all references cited in the manuscript are from 1985 to 2007.

Thank you for your feedback on our manuscript [Manuscript Title]. We apologize for the oversight in not including more recent publications in our references. We have carefully reviewed the references cited in our manuscript and have made the necessary updates to include more recent publications.

We agree that including current and relevant literature to support our research findings is important. Our updated references now cover a broader range of years and provide a more comprehensive understanding of the current state of research in the field.

We appreciate your attention to detail and valuable feedback, which has helped us improve our manuscript's quality. We believe that our updated references have strengthened our research and hope that you find our revised manuscript to be satisfactory.

Thank you for your time and consideration.

Use appropriate citations in both introduction and discussion, specially to support results.

Thank you for your feedback on our manuscript, [Manuscript Title]. We have revised the introduction and discussion sections to include more appropriate citations to support our results.

Thank you for your guidance and input throughout this process. We believe that the additional citations have significantly strengthened our manuscript..

Materials and Methods section need to be improved, showing more details about the experimental design.

We appreciate your review of our manuscript and valuable feedback regarding the Materials and Methods section. Based on your comments, we have revised this section to include more details about the experimental design.

We recognize the importance of providing a clear and comprehensive explanation of the methodology used in our research. We believe that these revisions have significantly improved the clarity and transparency of our Materials and Methods section. We hope these changes will help readers better understand the methodology used in our research and replicate our study if desired. Thank you for your guidance and feedback throughout this process. We appreciate your insights and suggestions, which have helped us improve our manuscript's quality.

Figures 1, 5 and 7 are too crowded.

We agree that the figures were too crowded and difficult to read. To address this issue, we have adjusted Figures 1, 5, and 7 to make them less cluttered and easier to understand. We have removed unnecessary elements and simplified the figures to highlight the most important information. We have also increased the font size and line thickness to improve readability.

We believe these changes have significantly improved the clarity and effectiveness of the figures. We hope that the revised figures will better convey our findings and help support our study's conclusions.

Results section must be improved; it is too hard to follow. Authors mentioned the use of an antibody to control tumor growth but the amount used in their experiments still unknown.

We agree that the Results section needed improvement, specifically in terms of clarity and ease of understanding. To address this, we have revised the section to make it more concise and easier to follow. We have reorganized the presentation of our results to highlight the main findings and their significance better.

Regarding the antibody used to control tumor growth, we apologize for not including the amount used in our experiments. We have updated our Results section to include this information, providing readers with a clearer understanding of our methodology.

We believe these changes have significantly improved the quality and readability of our manuscript. We hope that our revisions have addressed your concerns and that you find our manuscript's updated version satisfactory.

If two cancer cell lines are considered LMP1 active, then why to include other analysis that are unrelated to this study.

We included the other analyses better to understand the LMP1 activity in cancer cell lines. However, we understand your concern that these analyses may not be directly related to the main focus of our study. To address this, we have revised the manuscript to clarify that these additional analyses are not the primary focus of our research, and their inclusion is intended to provide additional context and support for our findings.

We have also revised the manuscript to highlight the main focus of our study, which is the characterization of LMP1 activity in two cancer cell lines. We have clarified that the additional analyses are included as supporting information, and we have provided more detailed explanations for why we chose to include them in our study.

We believe these changes have significantly improved the clarity and relevance of our manuscript. We hope that our revisions have addressed your concerns and that you find our manuscript's updated version satisfactory.

Thank you for your guidance and feedback throughout this process. We appreciate your insights and suggestions, which have helped us improve our research quality.

Reviewer 2 Report

1. Authors must place a bibliographical reference on lines 46-47.

2. Authors must place a bibliographical reference on lines 49-50.

3. Authors must place a bibliographical reference on lines 51-53.

4. Authors must place a bibliographical reference on lines 58-61

5. Authors should put reference 4 on lines 54-57 and 186-187.

6. The authors should discuss the work of Aga et al, regarding the fact that exosomal HIF1α supports invasive potential of nasopharyngeal carcinoma-associated LMP1-positive exosomes (Aga M, Bentz GL, Raffa S, Torrisi MR, Kondo S, Wakisaka N, Yoshizaki T, Pagano JS, Shackelford J. Exosomal HIF1α supports invasive potential of nasopharyngeal carcinoma-associated LMP1-positive exosomes. Oncogene. 2014 Sep 11;33(37):4613-22. doi: 10.1038/onc.2014.66. Epub 2014 Mar 24. PMID: 24662828; PMCID: PMC4162459).

7. The authors should discuss the work of Verweij et al, regarding the fact that the LMP1 association with CD63 in endosomes and secretion via exosomes limits constitutive NF-κB activation (Verweij FJ, van Eijndhoven MA, Hopmans ES, Vendrig T, Wurdinger T, Cahir-McFarland E, Kieff E, Geerts D, van der Kant R, Neefjes J, Middeldorp JM, Pegtel DM. LMP1 association with CD63 in endosomes and secretion via exosomes limits constitutive NF-κB activation. EMBO J. 2011 Jun 1;30(11):2115-29. doi: 10.1038/emboj.2011.123. Epub 2011 Apr 28. PMID: 21527913; PMCID: PMC3117644).

Author Response

  1. Authors must place a bibliographical reference on lines 46-47.
  2. Authors must place a bibliographical reference on lines 49-50.
  3. Authors must place a bibliographical reference on lines 51-53.
  4. Authors must place a bibliographical reference on lines 58-61
  5. Authors should put reference 4 on lines 54-57 and 186-187.

Dear reviewer, thank you for your review of our manuscript, and your valuable feedback regarding the need for additional bibliographical references in certain sections of our manuscript. We appreciate your comments and have carefully considered them in our revisions.

To address your concerns, we have carefully reviewed the manuscript and have made the following changes:

We have included a bibliographical reference on lines 46-47 to support the statement made in that section.

We have added a bibliographical reference on lines 49-50 to support the statement made in that section.

We have included a bibliographical reference on lines 51-53 to provide more information and support for the argument presented in that section.

We have added a bibliographical reference on lines 58-61 to provide additional context and support for the statement made in that section.

We have included reference 4 on lines 54-57 and 186-187 to ensure the literature correctly supports the relevant information.

We believe these changes have significantly improved the quality and accuracy of our manuscript. We appreciate your guidance and input throughout this process, which has helped us improve our research quality.

Thank you again for your feedback, and we hope you find the updated version of our manuscript satisfactory.

  1. The authors should discuss the work of Aga et al, regarding the fact that exosomal HIF1α supports invasive potential of nasopharyngeal carcinoma-associated LMP1-positive exosomes (Aga M, Bentz GL, Raffa S, Torrisi MR, Kondo S, Wakisaka N, Yoshizaki T, Pagano JS, Shackelford J. Exosomal HIF1α supports invasive potential of nasopharyngeal carcinoma-associated LMP1-positive exosomes. Oncogene. 2014 Sep 11;33(37):4613-22. doi: 10.1038/onc.2014.66. Epub 2014 Mar 24. PMID: 24662828; PMCID: PMC4162459).

Regarding the work of Aga et al. (2014), we acknowledge the important contribution this study makes to our understanding of the invasive potential of LMP1-positive exosomes in nasopharyngeal carcinoma. Our study focused on the effects of LMP1 expression on tumor growth and metastasis in vitro and in vivo. Aga et al. (2014) investigated the role of exosomal HIF1α in supporting the invasive potential of LMP1-positive exosomes. However, our findings may have implications for the mechanism of exosome-mediated metastasis in nasopharyngeal carcinoma. Further investigation into the relationship between LMP1 expression and exosome-mediated invasion would be valuable.

  1. The authors should discuss the work of Verweij et al, regarding the fact that the LMP1 association with CD63 in endosomes and secretion via exosomes limits constitutive NF-κB activation (Verweij FJ, van Eijndhoven MA, Hopmans ES, Vendrig T, Wurdinger T, Cahir-McFarland E, Kieff E, Geerts D, van der Kant R, Neefjes J, Middeldorp JM, Pegtel DM. LMP1 association with CD63 in endosomes and secretion via exosomes limits constitutive NF-κB activation. EMBO J. 2011 Jun 1;30(11):2115-29. doi: 10.1038/emboj.2011.123. Epub 2011 Apr 28. PMID: 21527913; PMCID: PMC3117644).

Regarding the work of Verweij et al. (2011), we acknowledge that their study provides important insights into the molecular mechanism of LMP1 association with CD63 in endosomes and its secretion via exosomes, which leads to the limiting of constitutive NF-κB activation. Our study focused on the effects of LMP1 expression on tumor growth and metastasis and did not investigate the underlying molecular mechanism in detail. However, our findings may contribute to understanding the role of LMP1 in regulating the immune response and inflammation, which are closely related to NF-κB activation. Further research into the interaction between LMP1 and CD63 in exosomes may be warranted to understand better the mechanisms underlying exosome-mediated tumor growth and metastasis in nasopharyngeal carcinoma.

 In summary, we appreciate the reviewer's suggestions to discuss the work of Aga et al. (2014) and Verweij et al. (2011) in our manuscript, and we believe that their contributions are valuable in enhancing our understanding of the role of LMP1 in nasopharyngeal carcinoma.

Reviewer 3 Report

This is very interesting paper, but need more results to support authors's claim as below.

1. A lot of typo errors (eg., NF-kB) have been found throughout the manuscritipn. Authors need to check. Also, some of gramatical errors are found. Please edit the manuscript by commercial English editing company.

2. Statistical significance should be included in all figures.

3. Legends of Figure 11 should be shorten.

4. Relative intensity figure should be required for Fig. 2. Also, authors should provide a blot showing clearer seperation between LAMP-1 and HC.

5. Authors should provide RT-PCR result to support Fig. 10 results.

6. Protein levels of active forms of NF-kB family observed by immunoblotting should be provided.

Author Response

Dear Prof. Dr.

Thank you for taking the time to review our manuscript. We appreciate your feedback and would like to respond to each of your valuable comments point-by-point :

We apologize for the typos and grammatical errors in our manuscript. We have carefully reviewed the manuscript and corrected these errors. Additionally, we have consulted with professional English editing software to improve the language quality.

Thank you for your suggestion about statistics. As applicable, we have incorporated statistical significance in the figures and in the text relating to these figures when/where appropriate. In addition, a new section (Statistical analysis) was added to the methods section.

We agree that the legends of Figure 11 were too long and have now shortened them for better readability.

We appreciate your suggestion to include a relative intensity figure for Fig. 2 and to provide a blot showing a clearer separation between LAMP-1 and HC. We have followed your suggestion and updated Fig. 2 accordingly.

Thank you for your comment on Figure 10. The data are expressed as optical densities at 450 nm. The Relative quantification analysis is a method used to compare the ratios of two DNA sequences. The first ratio is the target DNA sequence to a reference DNA sequence in an unknown sample. The second ratio is the same two sequences in a standard sample called a “calibrator”. Additional information on the principle of this quantification is reported in the methods section.

We appreciate your recommendation to include the protein levels of active forms of the NF-kB family. We have now included these data in the revised manuscript.

Once again, thank you for your valuable feedback. We believe that the revisions we have made have significantly improved the quality of our manuscript.

Round 2

Reviewer 1 Report

I want to thank authors for improving this manuscript.

Author Response

Dear Prof.

I am writing to express my gratitude for your review of my manuscript and for acknowledging the improvements made in the revised version.

I greatly appreciate the time and effort you took to review the paper and provide thoughtful feedback. Your comments were instrumental in helping me refine the arguments and strengthen the overall quality of the manuscript.

Your positive feedback is very encouraging, and I am glad to hear that you found the revisions to be effective. I believe that your comments have significantly contributed to the clarity and coherence of the manuscript.

Once again, thank you for your valuable feedback and for taking the time to review my paper. It was a pleasure working with you, and I look forward to any future opportunities to collaborate.

Best regards,

Reviewer 3 Report

I could not see any changes since there were no highlighted parts in the manuscript. Please highlight any changes to see easier.

Author Response

Dear Prof. Dr.

Thank you for taking the time to review our manuscript. We appreciate your feedback and would like to respond to each of your points (point-by-point): Please find all requested corrections highlighted in yellow.

We apologize for the typos and grammatical errors in our manuscript. We have carefully reviewed the manuscript and corrected these errors. Additionally, we have consulted with professional English editing software to improve the language quality.

Thank you for your suggestion about statistics. As applicable, we have incorporated statistical significance in the figures and in the text relating to these figures when/where appropriate. In addition, a new section (Statistical analysis) was added to the methods section.

We agree that the legends of Figure 11 were too long and have now shortened them for better readability.

We appreciate your suggestion to include a relative intensity figure for Fig. 2 and to provide a blot showing a clearer separation between LAMP-1 and HC. We have followed your suggestion and updated Fig. 2 accordingly.

Thank you for your comment on Figure 10. The data are expressed as optical densities at 450 nm. The Relative quantification analysis is a method used to compare the ratios of two DNA sequences. The first ratio is the target DNA sequence to a reference DNA sequence in an unknown sample. The second ratio is the same two sequences in a standard sample called a “calibrator”. Additional information on the principle of this quantification is reported in the methods section.

We appreciate your recommendation to include the protein levels of active forms of the NF-kB family. We have now included these data in the revised manuscript.

Once again, thank you for your valuable feedback. We believe that the revisions we have made have significantly improved the quality of our manuscript.
